# Gaussian-gated LSTM: Improved convergence by reducing state updates

## Abstract

Recurrent neural networks can be difficult to train on long sequence data due to the well-known vanishing gradient problem. Some architectures incorporate methods to reduce RNN state updates, thereby allowing the network to preserve memory over long temporal intervals. We propose a timing-gated LSTM RNN model, called the Gaussian-gated LSTM (g-LSTM) for reducing state updates. The time gate controls when a neuron can be updated during training, enabling longer memory persistence and better error-gradient flow. This model captures long temporal dependencies better than an LSTM on very long sequence tasks and the time gate parameters can be learned even from a non-optimal initialization. Because the time gate limits the updates of the neuron state, the number of computes needed for the network update is also reduced. By adding a computational budget term to the training loss, we obtain a network which further reduces the number of computes by at least $10\times$. Finally, we propose a temporal curriculum learning schedule for the g-LSTM that helps speed up the convergence time of the equivalent LSTM on long sequences.

## 1 Introduction

Numerous methods and architectures have been proposed to mitigate the vanishing gradient problem in RNNs, with LSTMs (Hochreiter & Schmidhuber, 1997) as one of the first prominent solutions doing so by including gating structures in the computation. Although the LSTM has excelled at handling many tasks (Schmidhuber, 2015; Lipton, 2015), it still has difficulties in learning complex and long time dependencies (Neil et al., 2016; Chang et al., 2017; Trinh et al., 2018).

In the last few years, various methods which reduce the state updates of an RNN (LSTM) have been explored to better learn long time dependencies from data. Clockwork RNNs (Koutnik et al., 2014) group the hidden units of the RNN into "modules," where each module is executed at pre-specified time steps thereby skipping time steps which helps learn longer time dependencies. Recently, various other methods have been proposed which can be characterized by the use of additional "time gates," $k_n$, that control the information flow from one time step to the next (Krueger et al., 2016; Campos et al., 2017). Phased LSTM (PLSTM) (Neil et al., 2016) learns a parameterized function, $k_n$, from the the time input of the current state and was proven to be successful at learning over long sequences.

The PLSTM time gate, parameterized by period, phase, and ratio parameters for each hidden unit, is defined through a modulo function with an ill-defined gradient. Furthermore, with periodic functions being hard to learn using gradient-based methods (Shamir, 2016) and with $k_n$ being periodic, the PLSTM was unable to learn the time gate parameters and hence relied on careful initialization. In order to offset these difficulties, this work proposes a new LSTM variant called the Gaussian-gated LSTM (g-LSTM). Similar to the PLSTM, it is an LSTM model with a parameterized $k_n$ but with only two parameters per hidden unit. Unlike the PLSTM which uses a periodic formulation for $k_n$, the g-LSTM uses a Gaussian function.

We show in this work that the g-LSTM can provide a number of possible advantages over the LSTM, in particular, on long sequence tasks that pose convergence problems during training:

- The g-LSTM network can process very long sequences by reducing the time over which the neurons can be updated. It converges faster than the LSTM, especially on sequences that are over 500 steps.

- The "openness" of the neuron for an update can be adapted according to the task during training, even for extreme, non-optimal initializations of the time gate parameters.

- By introducing a computational budget term into the loss function during training, the "openness" of the neuron can be optimized for a reduced computational budget. This reduction can be achieved with little or no degradation to the network performance and is useful for network pruning.

- A "temporal curriculum" training schedule can be set up for the g-LSTM so that it helps to speed up the convergence of a normal LSTM.

The paper is structured as follows: In section 2, we discuss briefly the related work. Then, in section 3, we present the formulation of the g-LSTM, the datasets used in this work and details about the experimental hyperparameters. In section 4, we present experiments demonstrating the usefulness of the g-LSTM with respect to the four claims listed above. We provide gradient analysis in section 5 to further explain the faster convergence results of the g-LSTM. Finally in section 6, we conclude with a brief discussion of the results.

## 2 RELATED WORK

There have been a multitude of proposed methods to improve the training of RNNs, especially for long sequences. Apart from incorporating additional gating structures, for example the LSTM and the GRU (Cho et al., 2014), more recently various techniques were proposed to further increase the capabilities of recurrent networks to learn on sequences of length over 1000. Proposed initialization techniques such as the orthogonal initialization of kernel matrices (Cooijmans et al., 2016), *chrono initialization* of the biases (Tallec & Ollivier, 2018), and diagonal recurrent kernel matrices (*e.g.* Li et al. (2018)) have demonstrated success. Trinh et al. (2018) propose using truncated backpropagation with an additional auxiliary loss to reconstruct previous events.

Methods that enable more efficient learning on long temporal sequences use solutions that preserve memory over longer timescales. Such solutions were first explored by Koutnik et al. (2014) in the Clockwork RNN (CW-RNN). This network skips state updates by allowing different neurons to be "activated" on different, modulated clock cycles. More recently proposed models for skipping updates include the Phased LSTM (PLSTM) (Neil et al., 2016) which uses a modulo-periodic timing gate to limit state updates; the Zoneout network (Krueger et al., 2016) which skips state updates in a random manner; and the Skip RNN (Campos et al., 2017) which learns a state skipping scheme from the data to shorten the effective sequence length for the task. Additionally, the LSTM-Jump (Yu et al., 2017) uses a reinforcement learning algorithm to learn when to skip state updates, showing a method to more quickly process (long) sequential data with an RNN while maintaining an accuracy comparable to a baseline LSTM.

It has been suggested but not yet demonstrated in the literature that the parameters of the CW-RNN clock cycle and PLSTM timing gate could be learned in training. Currently, the implementation of these networks requires a careful initialization of these parameters. With the Gaussian-gated LSTM (g-LSTM) in this work we present a time gated RNN network that converges on long sequence tasks and also has the ability to learn its time gate parameters even when initialized in a nonoptimal way.

## 3 METHODS

### 3.1 G-LSTM

The g-LSTM is an LSTM model with an additional time gate (Fig. 1). This time gate is used to regulate the information flow in time. Equations 1 - 3 describe the update equations for the hidden and cell states of the LSTM. Equations 4 and 5 describe the gating mechanism of the time gate, $k_n$.

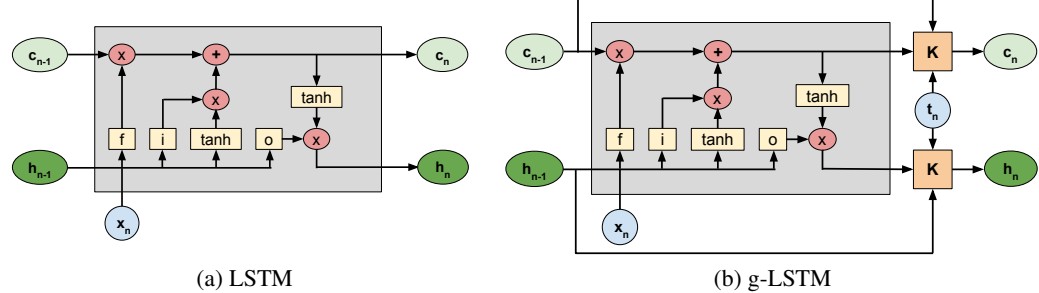

(a) LSTM

(b) g-LSTM

Figure 1: Comparison of the LSTM and g-LSTM models. K is the computational block for the time gate in (b).

$$i_n = \sigma(x_n W_{xi} + h_{n-1} W_{hi} + b_i), f_n = \sigma(x_n W_{xf} + h_{n-1} W_{hf} + b_f) \tag{1}$$

$$\tilde{c}_n = f_n \odot c_{n-1} + i_n \odot \sigma(x_n W_{xg} + h_{n-1} W_{hg} + b_g) \tag{2}$$

$$o_n = \sigma(x_n W_{xo} + h_{n-1} W_{ho} + b_o), \tilde{h}_n = o_n \odot \tanh(\tilde{c}_n) \tag{3}$$

$$c_n = k_n \odot \tilde{c}_n + (1 - k_n) \odot c_{n-1} \tag{4}$$

$$h_n = k_n \odot \tilde{h}_n + (1 - k_n) \odot h_{n-1} \tag{5}$$

In a standard LSTM, the gating functions $i_n$, $f_n$, $o_n$, represent the *input*, *forget*, and *output* gates respectively at sequence index $n$. $c_n$ is the cell activation vector, and $x_n$ and $h_n$ represent the input feature vector and the hidden output vector, respectively. The cell state $c_n$ is updated with a fraction of the previous cell state that is controlled by $f_n$, and a new input state created from the element-wise (Hadamard) product, denoted by $\odot$, of $i_n$ and the candidate cell activation as in Eq. 2.

In the g-LSTM, we further control the cell state and the output hidden state through the $k_n$ gate which is independent of the input data and hidden states, and is purely dependent on the time input corresponding to the sequence index $n$. The use of the Hadamard product ensures that each hidden unit is independently controlled by the corresponding time gate unit, thus enabling the different units in the layer to process the input at different time scales.

The time gate $k_n$ is defined based on a Gaussian function as: $k_n = e^{-(t_n - \mu)^2 / \sigma^2}$ where the mean parameter, $\mu$, defines the time when the hidden unit is "open" and the standard deviation, $\sigma$, controls the openness of the time gate for each unit around its corresponding $\mu$. The time inputs $\mathbf{t} = \{t_1, t_2, ..., t_n, ..., t_N\}$ for the sequence $\mathbf{x} = \{x_1, x_2, ..., x_n, ..., x_N\}$ can correspond to the physical notion of time at the respective sequence input. In the absence of a standard notion of time, we use the sequence indices as the time input, i.e. $\mathbf{t} = \{1, 2, ..., n, ..., N\}$. In this work, we assume this notion of time by default. The "openness" of $k_n$ for a neuron is defined by the parameterization of its Gaussian function.

### 3.2 BACK PROPAGATION FOR G-LSTM

An important characteristic of the g-LSTM is reduced gradient flow in back propagation training methods. By having the gating structure as in Eqs. 4 and 5 there are fewer gradient product terms, which reduces the likelihood of vanishing or exploding gradients. In a gradient descent learning scheme for a given loss function, $L$, when training the recurrent parameters, $W_h$ (from Eqs. 1 - 3), the gradient as in Eq. 6 is used.

$$\frac{\partial L}{\partial W_h} = \frac{\partial L}{\partial h_N} \frac{\partial h_N}{\partial W_h} \tag{6}$$

By the chain rule $\frac{\partial h_N}{\partial W_h}$ expands for all time steps of the sequence, $n \in \{1, ..., N\}$. Because each output state is gated by the time gate, $k_n$, the gradient terms in the expansion of $\frac{\partial h_N}{\partial W_h}$ are scaled by $k_n$. When the time gate is open less often, *i.e.* with a small $\sigma$ value, then there are fewer influential gradient terms. More details are given in Appendix A.

| Dataset | # units | Initialization | | Performance | |
| --- | --- | --- | --- | --- | --- |
| | | $\mu$ | $\sigma$ | g-LSTM | LSTM |
| Adding (N=1000) | 110 | $\sim U(300, 700)$ | 40 | $3.8 \cdot 10^{-5}$ | $1.4 \cdot 10^{-3}$ |
| Adding (N=2000) | 110 | $\sim U(500, 1500)$ | 40 | $1.3 \cdot 10^{-3}$ | $1.6 \cdot 10^{-1}$ |
| sMNIST | 110 | $\sim U(1, 784)$ | 250 | 1.3% | 1.8% |
| sCIFAR-10 | 128 | $\sim U(1, 1024)$ | 650 | 41.1% | 41.8% |

Table 1: Network architectures and performance for the convergence experiments in subsection 4.1. The performance metric is the final mean squared error (MSE) loss for the adding task, and the label error rate for both sMNIST and sCIFAR-10.

## 3.3 DATASETS

The experiments described in the paper are carried out on the adding task and two standard long sequence datasets: the sequential MNIST and the sequential CIFAR-10 datasets.

**Adding task:** In order to test the long sequence learning capability of the g-LSTM, we use the adding task (Hochreiter & Schmidhuber, 1997). In this task, the network is presented with two sequences of length $N$, $\mathbf{x} = \{x_1, ..., x_N\}$, $x_n \sim U(0, 1))$ and $\mathbf{m} = \{m_1, ..., m_N\}, m_n \in \{0, 1\}, \sum_{t=1}^{N} m_n = 2$. The sequence $\mathbf{m}$ has exactly two values of 1 and the remaining values of the sequence are 0. The indices of the "1" values are chosen at random. For each pair in the sequence $(\mathbf{x}, \mathbf{m})$, the associated label value, $y$, is the sum of the two values in $\mathbf{x}$ corresponding to the "1" values of $\mathbf{m}$. The objective of this task is to minimize the mean squared error between the predicted sum from the network, $\hat{y}$, and the labeled sum, $y$. A new training set of 5000 sequence samples is presented in every epoch during training in order to avoid overfitting. The test set consists of a separate 5000 samples. For $N > 1000$, it is known that LSTMs have difficulty learning the task and hence we focus on values of $N > 1000$ in this work.

**sMNIST:** The sequential MNIST dataset is widely used to analyze the performance of a recurrent model. This dataset consists of 60,000 training samples and 10,000 test samples, each a single vector sequence of length 784 corresponding to the $28 \times 28$ pixel images in the MNIST dataset (LeCun et al., 1998). We also use permuted MNIST (pMNIST), a permuted variant of the sMNIST dataset where the sequences are processed with a fixed random permutation, making the task harder.

**sCIFAR-10:** The sequential CIFAR-10 dataset is another long sequence dataset based on CIFAR-10 (Krizhevsky et al., 2014) with 10 classes. The $32 \times 32$ RGB pixel images are reshaped into sequences of length 1024 with 3 dimensional features corresponding the RGB channels at every time step. Like in the sMNIST dataset, the dataset consists of 60,000 training samples and 10,000 test samples.

## 3.4 EXPERIMENTAL HYPERPARAMETERS

For the adding task, a mean squared error (MSE) loss was used with the Adam optimizer (Kingma & Ba, 2014) with a learning rate of $10^{-3}$. The g-LSTM time gate parameters were trained using a learning rate of $10^0$. For both sMNIST and sCIFAR-10 datasets, the cross-entropy loss function was used along with the RMSProp optimizer (Tieleman & Hinton, 2012) with a learning rate of $10^{-3}$. Decay parameters of 0.5 and 0.9 were used for sMNIST and sCIFAR-10, respectively. The bias of the forget gate is initialized to 1 following (Jozefowicz et al., 2015).

## 4 RESULTS

Section 4.1 presents results that demonstrate the faster convergence properties of the g-LSTM on long sequence tasks. Section 4.2 shows the trainability of the time gate parameters of the g-LSTM even when the parameters are initialized in a non-optimal way. Section 4.3 presents a modified loss function used during training to reduce the number of computes for the network update and Section 4.4 presents a new "temporal curriculum" learning schedule that allows g-LSTMs to help LSTMs converge faster.

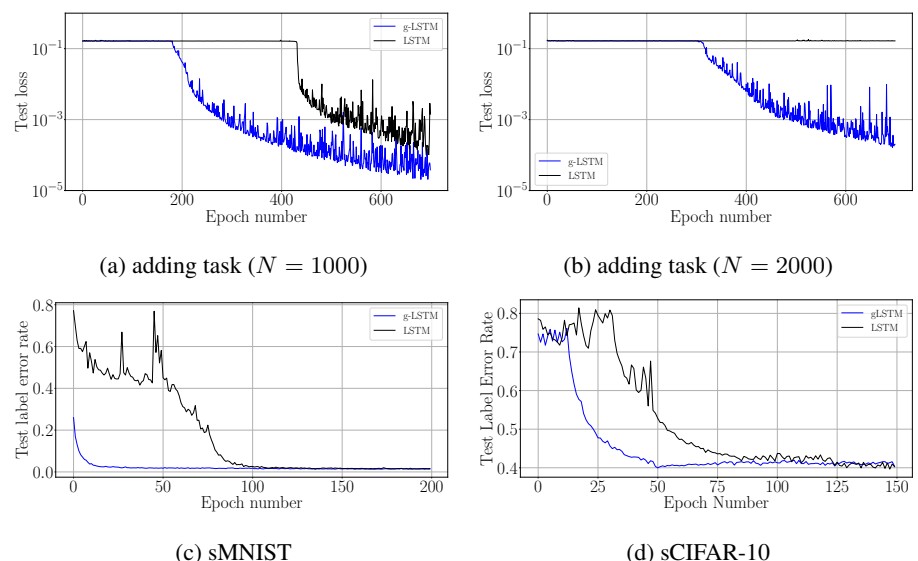

Figure 2: Test loss and test label error rates across training epochs for LSTM (black) and g-LSTM (blue) networks on different tasks.

## 4.1 FAST CONVERGENCE PROPERTIES OF G-LSTM

First, we look at the convergence properties of the g-LSTM on the long-sequence adding task, the sMNIST task and the sCIFAR-10 task. Table 1, above, details the network architectures used in the experiments in this section. Similar to the architecture from Trinh et al. (2018), the recurrent layer of the sCIFAR-10 network is followed by two 256 unit fully-connected (FC) layers, where Drop-Connect (Wan et al. (2013)) ($p = 0.5$) is applied to the second FC layer. The kernel matrices in the LSTM networks were initialized in an orthogonal manner as described in (Cooijmans et al., 2016).

The test performances of these networks during the course of the training on different datasets are shown in Fig. 2, while the corresponding final performance metrics at the end of training are shown in Table 1. From Fig. 2, it is evident that the test loss of the g-LSTM decreases faster in training than the LSTM across all datasets. Further experiments show that this trend is maintained with different training optimizers, LSTM initializations including the bias initialization following Tallec & Ollivier (2018), and network sizes as shown in Appendix D.

Table 2 compares the performance of various networks including the g-LSTM and the baseline LSTM on sMNIST and sCIFAR-10 (from Table 1). The results show that the g-LSTM consistently performs better than the LSTM and has a similar performance to other state-of-the-art networks. Different network sizes were also investigated for the sMNIST task, see Appendix D.

| Network | sMNIST | pMNIST | sCIFAR-10 |
|---|---|---|---|
| g-LSTM (ours) | 1.3% | 7.5% | 41.1% |
| LSTM (ours) | 1.8% | 8.4% | 41.8% |
| r-LSTM (Trinh et al., 2018) | 1.6% | 4.8% | 27.8% |
| Zoneout (Krueger et al., 2016) | 1.3% | 6.9% | - |
| IndRNN (6 layers) (Li et al., 2018) | 1.0% | 4.0% | - |
| BN-LSTM (Cooijmans et al., 2016) | 1.0% | 4.6% | - |
| Skip LSTM (Campos et al., 2017) | 2.7% | - | - |

Table 2: Comparison of label error rates across different networks.

### 4.2 TRAINABILITY OF THE TIME GATE PARAMETERS OF G-LSTM

To demonstrate that the g-LSTM can be trained even with non-optimal initializations, we look at the performance of the g-LSTM on the adding task with different time gate parameter initializations. We concern ourselves with sequences of length 1000 that are difficult for the LSTM. The time gate parameters are initialized in a way to temporally constrain the network so that it can only process for a short period of time. For example, a network with time gate parameters initialized with $\mu = 500$ and $\sigma = 40$ as in Figure 3 (a) can only process a short period of time around the middle of sequence. It follows that the network would be unable to learn with these parameters because in the adding task the input data is distributed equally across the sequence length ($T = 1000$). Therefore, in order to learn the task from this initialization, the time gate parameters must learn a distribution such that the gates over all hidden units are open across the entirety of the sequence.

We observe that the time gate parameters do learn, as shown in Figure 3 (b), thereby enabling the network to solve the task. Independent of various time gate initializations, the network reaches an MSE of around $3.9 \times 10^{-5}$ at the end of 700 epochs; details of which could be found in Appendix B. The ability of the network to learn the time gate parameters necessary to cover the entire sequence is especially significant because it shows that even with this narrow time window initialization that requires learning of the time gates, the g-LSTM learns the task, whereas the PLSTM does not learn the task as well. An example of this is shown in Appendix C.

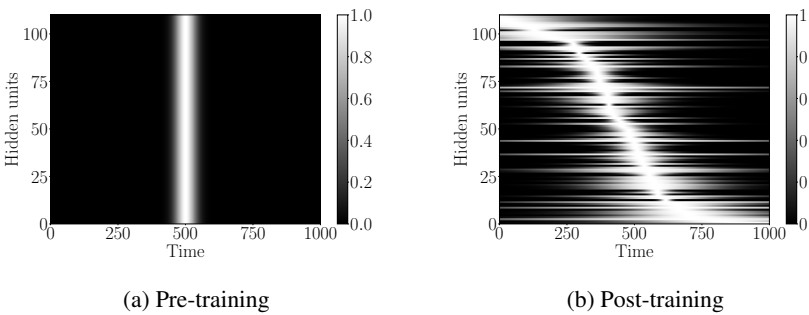

(a) Pre-training                    (b) Post-training

Figure 3: Time gate behavior pre and post training, demonstrating the ability of the network to learn from extreme initialization parameters. Here, $k_n$ is plotted as a function of time ($x$-axis) with black values corresponding to a fully closed gate (value 0) and white values corresponding to a fully open gate (value 1). Note that lower values of $\sigma$ ensure that the unit is processed only if the time input is $\mu$, while higher $\sigma$ values lead to the unit processed like in the standard LSTM, at all times.

### 4.3 REDUCTION IN COMPUTATION

Although the formulation of the g-LSTM appears to require more computes, it offers substantial speedup as a large proportion of the neurons can be skipped in a timestep at runtime. We can set a threshold on the time gate so that we skip all corresponding computations for time steps where $k_n$ is below this threshold. To further reduce the number of operations, it is preferred that the $\sigma$ of the $k_n$ for different neurons should be small but the network performance should not be significantly degraded. To achieve this goal, we included a "computational budget" loss term during the optimization of the gate parameters, $\mu$ and $\sigma$. The loss equation for updating the $k_n$ parameters is given by:

$$L = L_{data} + \lambda L_{budget}.$$

Similar to the Skip RNN network (Campos et al., 2017), a budget loss term which minimizes the average openness of the time gate over time is applied:

$$L_{budget} = \mathbb{E}[k_n] \approx \sum_{n=1}^{N} \sum_{j=1}^{J} k_n^{(j)}$$

for every neuron $j$ of the g-LSTM. The study was carried out on sMNIST using a network with 110 units, $\sigma$ initialized to 50, $\mu$ initialized uniformly at random between 1 to 784, and a $\lambda$ value of 1.

The network's performance of 2.2% LER was comparable to the network's performance of 1.3% when no additional budget constraint was imposed. The final $\sigma$ range for the budgeted g-LSTM is much smaller compared to that of the g-LSTM as shown in Fig. 4. There is only a slight increase in LER for the budgeted g-LSTM versus the g-LSTM (see Table 2), even though there is a significant decrease in the average time gate openness across all hidden units.

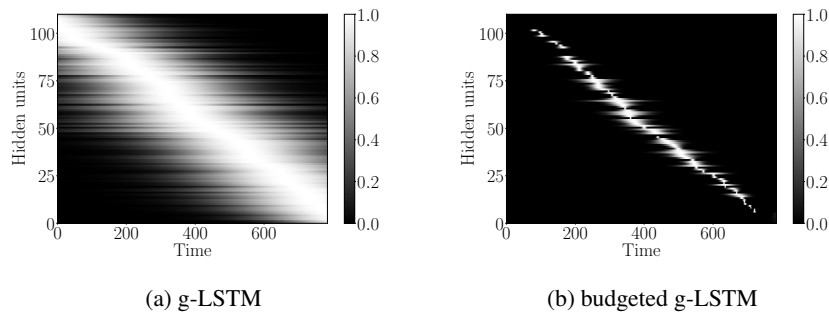

(a) g-LSTM                                (b) budgeted g-LSTM

Figure 4: Time gate behavior of (a) g-LSTM and (b) budgeted g-LSTM for 110 units post training.

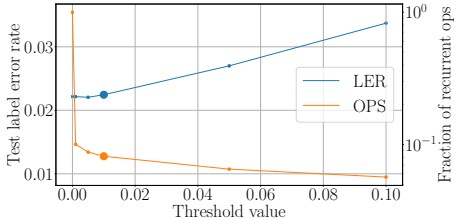
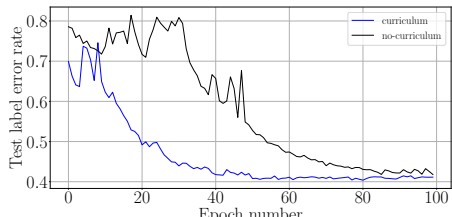

Figure 5: Reduction in computes as a function of threshold for budgeted g-LSTM.

Figure 6: Speed up in convergence of LSTM using the temporal curriculum learning schedule.

In order to reduce the number of computes, we set a threshold, $v_T$ for $k_n$ so that the update steps are carried out only if $k_n > v_T$, if $k_n < v_T$ the previous neuron state can be copied over to the current state. By increasing $v_T$, the number of computes decreases as shown in Fig. 5. In the case of $v_T = 0.01$, only 8.2% of the time gates are open on average across all hidden units and all time steps. Furthermore, the LER increased only slightly to 2.3% from 2.2%.

We give a quantitative estimate for the number of operations (Ops) corresponding to the number of update equations for a g-LSTM. In the estimate, we count a multiply and an add operation as 1 Op and non-linear functions as 5 Ops. For an LSTM, the number of operations is given by

$$N_{LSTM} = T H(8D + 8H + 29)$$

where $T$ is the number of time steps, $H$ is the number of hidden units, and $D$ is the dimension of the input data. For a g-LSTM, the number of operations is given by

$$N_{g-LSTM} = N_{LSTM} + N_{gate}$$

where $N_{gate} = 13 T H$ is the total number of operations for computing the time gate. The total number of operations for the g-LSTM network on the sMNIST dataset is around 80 MOps for $N = 110$ and $T = 784$, after thresholding the budgeted g-LSTM this number is reduced to 7.6 MOps. Additional $\lambda$ hyperparameters were also investigated for the sMNIST task, see Appendix D.

### 4.4 TEMPORAL CURRICULUM TRAINING SCHEDULE FOR LSTMS

We demonstrate that it is possible to train an LSTM network to converge faster on a difficult task by using a "temporal curriculum" training schedule for the equivalent g-LSTM network. According to this schedule, the initial $\sigma$ values of the g-LSTM network are increased continuously throughout the

training period ending up with high values by the end of training. With such high values, the time gates are essentially open, resulting in an LSTM network. At every training epoch, the lowest $\rho\%$ of the $\sigma$ values, $\hat{\sigma}$ in the layer are updated as: $\hat{\sigma} \longrightarrow (1 + \alpha) \cdot \hat{\sigma}$.

We analyze the impact of this training schedule for training an LSTM network on sCIFAR-10. For the equivalent g-LSTM network with 110 units, $\mu$ is initialized uniformly at random between 1 and 1024 and $\sigma$ is initialized to 50. An $\alpha$ value of 1/6 and $\rho$ value of 15% are chosen. To ensure that the time gate is fully open by the end of training, $\sigma$ is set to 5000 across all units during the last 10 epochs of training. The learning rate of the time gate parameters is set to 0, *i.e.* $\mu$ and $\sigma$ are no longer updated. Figure 6 shows that the temporal curriculum training schedule allows for faster convergence of an LSTM network. The final weights of the trained g-LSTM network can then be copied over to a LSTM network for inference.

## 5 GRADIENT FLOW

We present results regarding backpropagation flow through the LSTM and g-LSTM networks. Following the hypothesis presented in Section 3.2 on the reduced likelihood of vanishing or exploding gradients in the g-LSTM, we investigate the average gradient norms across time steps, similar to the work in (Krueger et al., 2016). We compute the gradient norms of the loss with respect to the hidden activations, the exact definition is given in Appendix E. Comparing the error propagation of the g-LSTM and LSTM networks on the SMNIST task (as in Sec. 4.1), Figure 7 shows the gradient norms at each time step after training for two different epochs.

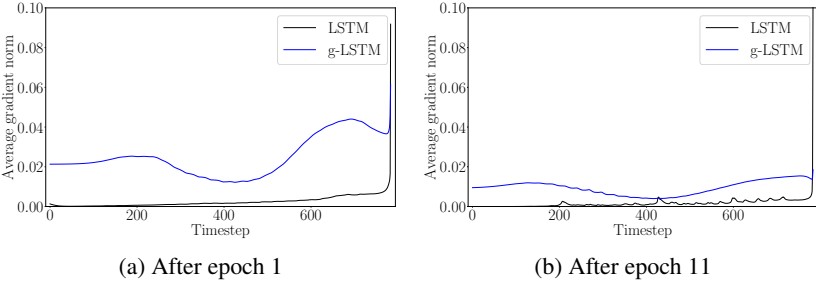

(a) After epoch 1        (b) After epoch 11

Figure 7: Average gradient norm through time steps for g-LSTM and LSTM.

Interpreting the gradient flow from higher to lower time steps (right to left), the gradients of the g-LSTM shown in Fig. 7 show higher gradient values in earlier time steps than the LSTM. It is possible that one of the reasons the g-LSTM converges more quickly (as in Fig. 2 (c)) is that this back-propagated gradient information is more consistent across time steps and does not vanish at early time steps.

## 6 CONCLUSION

This work proposes a novel RNN variant with a time gate which is parameterized by the input in time. The convergence speeds of the g-LSTM and LSTM are similar for short sequence tasks but the g-LSTM shows faster convergence and produces higher accuracies than LSTM networks on long sequence tasks, as demonstrated for adding task sequences which are longer than 1000 timesteps; and on the sMNIST and sCIFAR-10 datasets. We also demonstrate that the time gate parameters of the g-LSTM (unlike those of the PLSTM) are learnable even when the time gates are initialized in an extreme non-optimal manner for the adding task. The time gate of the g-LSTM can reduce the number of computes that is needed for the updates of the LSTM equations and with an additional loss term to reduce the compute budget, the $\sigma$ values of the time gate are reduced leading to a $10\times$ decrease in the number of actual computes and with little loss in network accuracy, for the sMNIST dataset. The observation that the budgeted g-LSTM has neurons which are closed by the timing gate suggests that this method can be used to prune a network. We also show that our proposed temporal curriculum training schedule for the g-LSTM can help a corresponding LSTM network to converge during training on long sequence tasks. For future work, it will be of interest to investigate whether these properties carry over to larger or domain-specific datasets.

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

## A    BACK PROPAGATION IN GAUSSIAN-GATED RNN

For ease of illustration we analyze the gradient of a plain RNN with a Gaussian time gate (Eqs. 7 and 8).

$$h_n = k_n \cdot \tilde{h}_n + (1 - k_n) \cdot h_{n-1} \tag{7}$$

$$\tilde{h}_n = f(W_x x_n + W_h h_{n-1}) \tag{8}$$

$$\frac{\partial h_N}{\partial W_h} = \frac{\partial \tilde{h}_0}{\partial W_h} \prod_{n=1}^{N} (k_n W_h f'_n + (1 - k_n)) + \sum_{n=1}^{N} (k_n f'_n h_{n-1}) \prod_{s=n+1}^{N} (k_s W_h f'_s + (1 - k_s)) \tag{9}$$

where $\frac{\partial \tilde{h}_0}{\partial W_h} = 1, h_0 = \tilde{h}_0 = 0$.

From Eq. 9 we can deduce some information about the advantages of the Gaussian time gate in gradient flow for two simple cases of the function $k_n$.

In Case 1 we choose a timing gate openness which corresponds to a very small $\sigma$ for the Gaussian gate, *i.e.* the gate is only open for 1 time step.

$$k_5 = 1, k_n = 0 \quad \forall \quad n \in \{1, ..., N\} \backslash \{5\}$$

$$\frac{\partial h_N}{\partial W_h} = f'_5 + f'_5 f'_5 h_4 = f'_5 W_h (1 + f'_5 h_4)$$

In Case 2 we choose a timing gate openness which corresponds to a slightly larger $\sigma$ for the Gaussian gate, *i.e.* it is open for 5 time steps.

$$k_2 = 1, k_3 = 1, k_4 = 1, k_5 = 1, k_6 = 1, k_n = 0 \quad \forall \quad n \in \{1, ..., N\} \backslash \{2, 3, 4, 5, 6\}$$

$$\frac{\partial h_N}{\partial W_h} = f'_2 W_h f'_3 W_h f'_4 W_h f'_5 W_h f'_6 W_h (1 + f'_2 h_1 + f'_3 h_2 + f'_4 h_3 + f'_5 h_4 + f'_6 h_5)$$

These cases show that there are fewer terms in the gradient for a timing gate that is opened for only a small fraction of the sequence.

## B COMPARING VARIOUS G-LSTM INITIALIZATIONS

|  | Initialization | | |
| Experiment ID | $\mu$ | $\sigma$ | Final MSE Loss |
| --- | --- | --- | --- |
| A1 | $\sim U(300, 700)$ | 1 | $4.4 \cdot 10^{-4}$ |
| A2 | $\sim U(0, 400)$ | 40 | $2.0 \cdot 10^{-5}$ |
| A3 | $\sim U(600, 1000)$ | 40 | $4.0 \cdot 10^{-4}$ |

Table 3: Adding task (T=1000): 110 unit g-LSTM network initializations and performances.

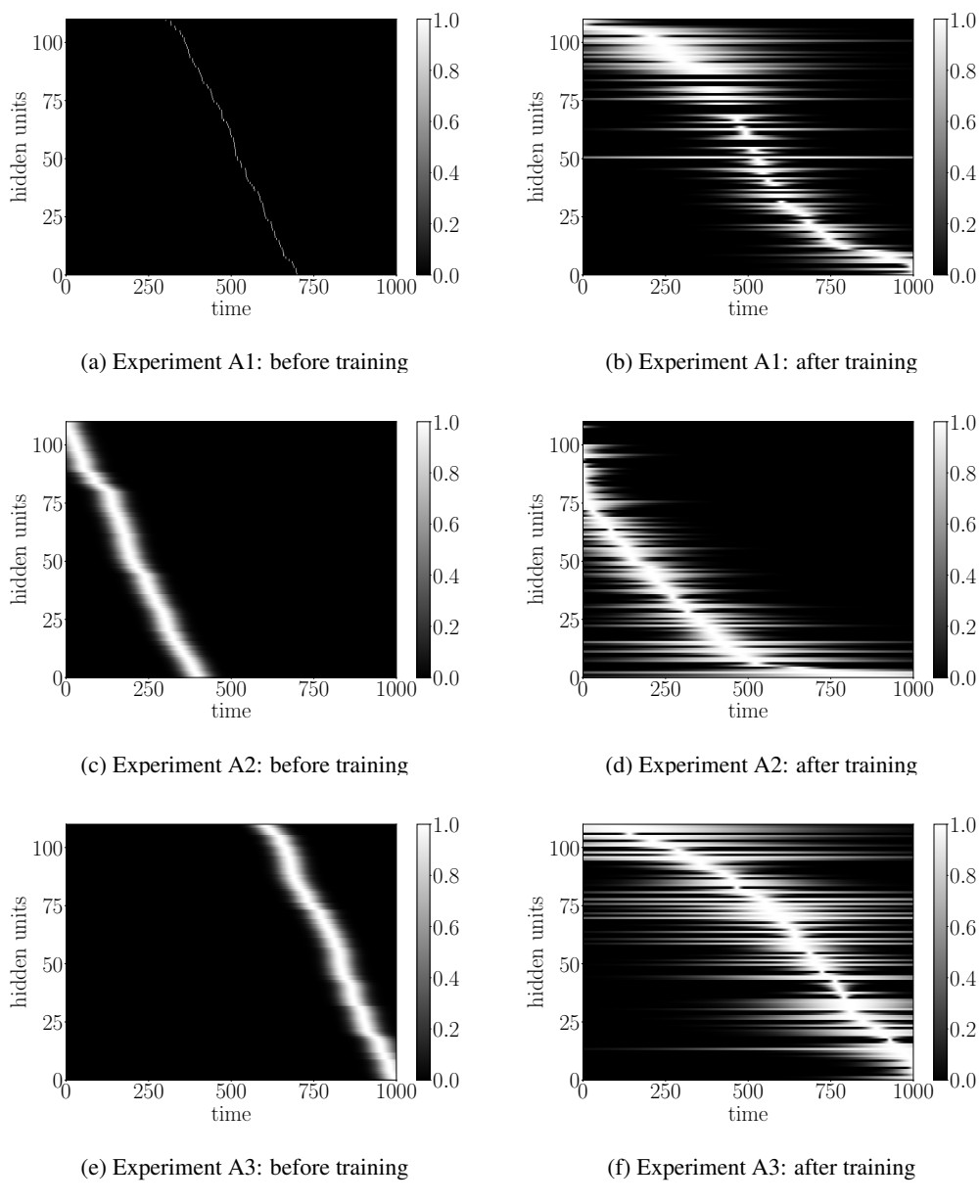

(a) Experiment A1: before training

(b) Experiment A1: after training

(c) Experiment A2: before training

(d) Experiment A2: after training

(e) Experiment A3: before training

(f) Experiment A3: after training

Figure 8: Additional experiments: timing gate openness for non-optimal initializations

## C COMPARING TIME GATE PARAMETER TRAINABILITY IN G-LSTM AND PLSTM

| Network | Initialization | Final MSE Loss |
|---------|----------------|----------------|
| g-LSTM | $\mu \sim U(300, 700), \sigma = 40$ | $7.7 \cdot 10^{-5}$ |
| PLSTM | $\tau = 1000, s \sim U(250, 650), r = 0.10$ | $2.4 \cdot 10^{-4}$ |

Table 4: Adding task (T=1000): Comparing 110 unit g-LSTM and PLSTM networks with similar initializations, MSE computed after training for 500 epochs.

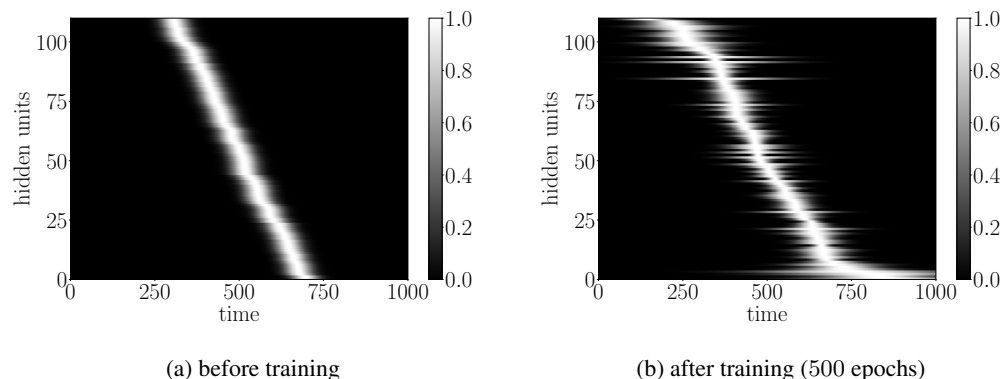

(a) before training        (b) after training (500 epochs)

Figure 9: g-LSTM

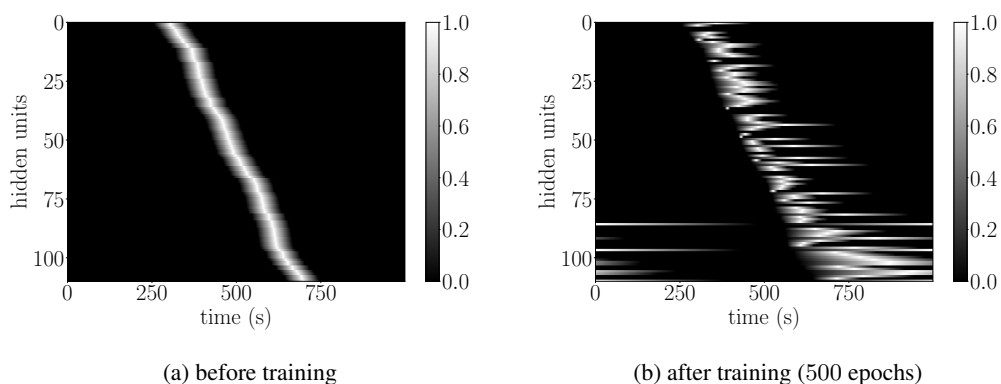

(a) before training        (b) after training (500 epochs)

Figure 10: PLSTM

# D    Hyperparameter Investigation

We look at the network performance for different hyperparameter values, focusing on the sMNIST task.

**Network Initialization and Optimizer**    In Fig. 2(c) of Section 4.1, we show that the g-LSTM network converges faster than the LSTM for the sMNIST task using the RMSProp optimizer and with an orthogonal initialization of LSTM kernels of both networks, as in (Cooijmans et al., 2016). In addition to using this initializer and optimizer we include results using the ADAM initializer (learning rate of $10^{-3}$) and a random weight initialization, "Xavier" as in (Glorot & Bengio, 2010). Across all of these different training techniques we consistently observe that the g-LSTM converges more quickly than the LSTM.

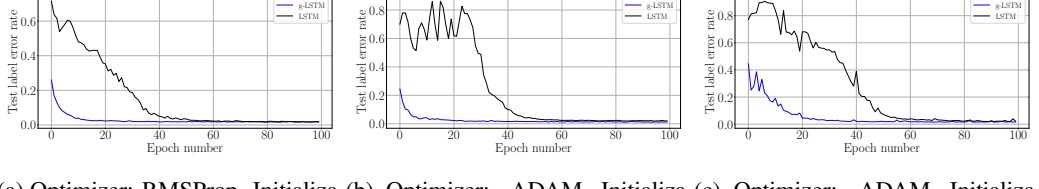

(a) Optimizer: RMSProp. Initializa-(b) Optimizer:  ADAM. Initializa-(c) Optimizer:  ADAM. Initializa-
tion: Xavier      tion: Xavier      tion: Orthogonal

Figure 11: Results for sMNIST with various optimizers and initializations.

We ran further experiments to compare the *chrono initialization* of the LSTM forget and input biases from (Tallec & Ollivier, 2018). The forget and input biases are set as $b_f \sim log(U([1, T_{max} - 1]))$ and $b_i = -b_f$ where $T_{max} = 784$ for the sMNIST task. The use of the time gate with the g-LSTM shortens the effective sequence length for each unit; to account for this, we also provide the results of using a smaller $T_{max}$ value for the *chrono initialization*, $T_{max} = \sigma = 250$. The comparison of both g-LSTM and LSTM with the *chrono initialization* and with the "constant initialization" ($b_f = 1$) in Fig. 12 shows that the g-LSTM with the constant initialization converges the fastest. We hypothesize that the g-LSTM can converge faster when using the constant initialization over the *chrono initialization* because the time gate's effect of sequence-length-shortening reduces the necessity for long memory, for which *chrono initialization* seeks to provide. We see that when we reduce the maximum temporal dependency for the *chrono initialization* (to $T_{max} = 250$, "chrono-g-LSTM-250") this g-LSTM network converges more quickly, similar to the g-LSTM with a constant bias initialization. This suggests that these two techniques, *chrono initialization* and a Gaussian time gate, could be used together to improve convergence in LSTM networks.

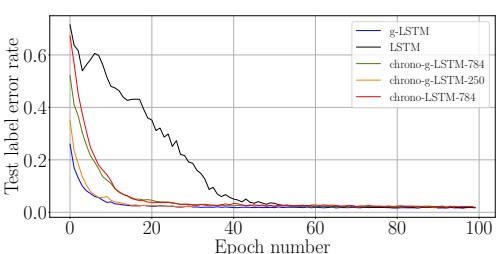

Figure 12: Comparative results on sMNIST with *chrono initialization* and constant initialization.

**Network Size**    Aside from the network size of 110 hidden units, we investigated the training convergence for two additional network sizes: 25 and 220 hidden units. Note that the LSTM for network size 25 is trained for 100 additional epochs until convergence was observed. Across all different network sizes the g-LSTM converges much faster than the LSTM network. With fewer hidden units,

as seen in Fig. 13 (a), an even more dramatic speed-up in convergence is seen for the g-LSTM compared with the LSTM. The final LERs (g-LSTM, LSTM) for each network size are: 25 units (2.79%, 3.58%), 110 units (1.35%, 1.81%) , 220 units (1.10%, 1.34%).

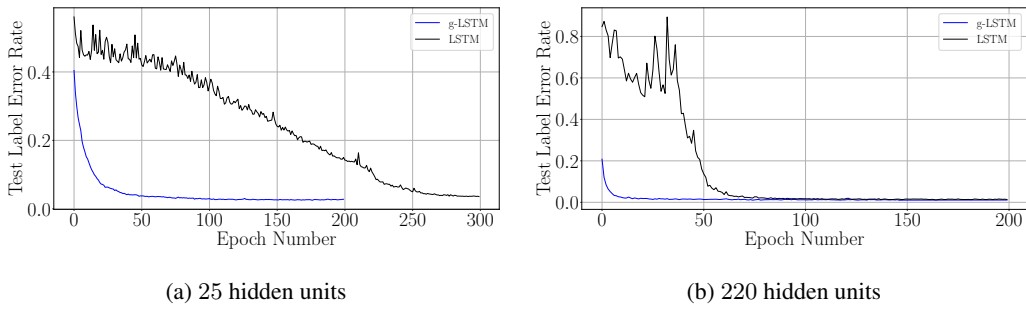

(a) 25 hidden units                        (b) 220 hidden units

Figure 13: Results for sMNIST using two different network sizes.

**Budgeted g-LSTM** We provide additional results of the budgeted network (subsection 4.3) for 2 additional $\lambda$ values, $\lambda = 0.1$ and $\lambda = 10$, comparing with the original result, for $\lambda = 1$. The final LERs: 2.4% ($\lambda = 0.1$), 2.2% ($\lambda = 1$), 2.8% ($\lambda = 10$). The number of computes used by the network trained with $\lambda = 10$ is significantly lower than the network that was trained for both $\lambda = 0.1$ and 1.

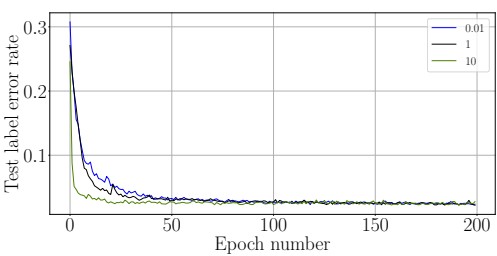

Figure 14: Label error rate on sMNIST for different $\lambda$ values (shown in legend).

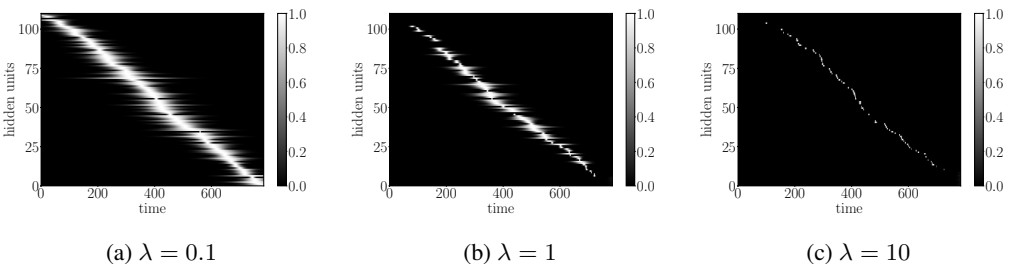

(a) $\lambda = 0.1$                  (b) $\lambda = 1$                  (c) $\lambda = 10$

Figure 15: Gate openness for different $\lambda$ values of the budgeted g-LSTM.

## E   AVERAGE GRADIENT NORM DEFINITION

The average gradient norm in Section 5 is defined as:

$$\Gamma \in \mathbb{R}_+^N$$

where $N$ is the number of time steps of the sequence (for SMNIST, $N = 784$).

$$\Gamma = \frac{1}{K} \mathbb{E}\left[\sum_{(k)} \left\| \frac{\partial L}{\partial h_n^{(k)}} \right\|\right] \approx \frac{1}{LK} \sum_{(l,k)} \left\| \frac{\partial L}{\partial h_n^{(k)}} \right\|$$

summing over all $L$ samples of the training set and all $K$ hidden units.

