# OpenReview forum: "Gaussian-gated LSTM: Improved convergence by reducing state updates"
_ICLR.cc/2019/Conference_

### Official Review · AnonReviewer1 · 2018-11-03
**Novel Contribution; Interesting speedup; too few self-critical; more evaluation needed**

**Rating:** 6
**Confidence:** 4

**Review:**


This paper focuses on the reduction of training time by various mechanisms. By introducing a time gate during training, it controls when a neuron (weights?) can be updated during training. By introducing and additional budget term in the loss function, training costs (number of computations) are reduced by one order of magnitude.
A major advantage of the newly introduced Gaussian-gated LSTM (g-LSTM; I suggest using a capital G for Gauss, e.g., GgLSTM).

Experiments are carried out on the adding-problem from 1997; the sequential MNIST and the sequential CIFAR-10 problem. In all experiments, g-LSTM converges faster. A few things would be of interest:
- clearly state the stopping criterium for training. Especially, I would still be interested to see, how Fig. 3d continues; it seems that the network begins to collapse (also a and be are interesting to see).
The "This work" in Table 2 is confusing; I would expect it to appear behind g-LSTM;
It appears that in the budgeted g-LSTM some units are not used at all (Figure 5b); Please comment on that.

In general, the paper makes the impression that it is overselling the contribution a bit too much. It would be nice to question the outcomes more and investigate the g-LSTM for the existence of possible problems which might be introduced by the omission of computations.

---

> ### Author Response · Authors · 2018-11-19
> **Response to AnonReviewer1**
>
> Thank you for your review.
>
> We have rewritten the abstract, introduction, and conclusion to highlight the specific  advantages  of  g-LSTM,  which  is  primarily  on  the  faster  convergence  than  the LSTM, on long sequence tasks.
>
> 1. We have added more experiments over a range of hyperparameters. These results are reported in Appendix D.
>
> 2. We have also added further evidence regarding a better gradient flow in the g-LSTM  network  compared  to  the  LSTM.  These  results  are  reported  in  a  new section (Section 5).
>
> 3. We have now emphasized in the main text (Introduction and Conclusion) that the advantages of g-LSTM over the LSTM are prominently on the faster convergence in tasks where the sequence length is over 500. For shorter sequences we see that both the gLSTM and LSTM networks perform similarly.
>
> 4. Regarding the stopping criterion, we run our experiments until convergence of all the models under consideration or until 700 epochs in the case of adding task.We have also updated the figures 3b and 3d to include more training epochs.
>
> 5. We have reordered the first two rows in table 2 to make clear that the figures reported for gLSTM and LSTM were based on our experiments.
>
> 6. Regarding the unused units in the budgeted g-LSTM, this could be because the network can solve without requiring all the 110 units as seen in the new added experiments in the Appendix D (with 25 units). We regard this as a feature which would be of help in pruning networks. We find this effect more prominent when increasing the 'lambda'  hyperparameter value, corresponding to the budget term in the loss function.

---

### Official Review · AnonReviewer2 · 2018-11-03
**Simple ideas but unconvincing results**

**Rating:** 5
**Confidence:** 4

**Review:**

The work takes inspiration from a recent work on phased LSTM, and proposes to add a Gaussian gate based on time to LSTM cells. With this additional gate, the network can skip updating the states by closing the time-gate, as a result enabling longer memory persistence, and better gradient flow. The authors also propose to add a budget term to force the time-gate to be closed most of the time as a way to save compute. Empirical results suggest the Gaussian-gated LSTMs perform better than regular LSTMs on tasks with long temporal dependencies. The authors also propose to use a curriculum training schedule in which the variance of the gaussian gates is continuously increased to speed up training of LSTMS.

pros:
1. The paper is clearly written and easy to follow;
2. The way the authors introduce time-dependent gating into LSTM is easy to follow and re-implement;
3. Experiments on various tasks of long temporal dependencies do show improvement over the standard LSTM cells;
4. The experiments on the adding task does show gLSTM is less sensitive to initialization than the phased LSTM;
5. The experiments on setting curriculum training schedule to improve convergence on LSTMs are insightful.

cons:
1. The work was framed as an easier-to-optimize alternative to the time-based gating mechanism introduced in phased LSTMs, which takes a parametrization form that is much harder to learn, the gating mechanism covered by the new model gLSTM however, is much more limited. The parametrization introduced in Phased LSTMs allows the memory cells and outputs of LSTMs to be updated periodically. gLSTMs on the other hand only allows updates within a single window over the entire sequence; As a result, one would expect phased LSTM to outperform gLSTM on tasks with periodical temporal dependencies;
2.  The empirical results are not convincing enough. gLSTM performs noticeably worse than several state-of-the-art work on improving long-term dependencies in RNNs. The authors did not give any explanation in the performance gap.

---

> ### Author Response · Authors · 2018-11-19
> **Response to AnonReviewer2**
>
> Thank you for your review.
>
> 1. It is possible that the PLSTM will outperform the g-LSTM on tasks which require a periodicity, but we have not found any datasets as yet where the PLSTM outperforms the g-LSTM. Moreover the time gate parameters in the g-LSTM network need not be carefully initialized, which is not true in the case of PLSTM.
>
> 2. The intention of Table 2 is not to do a direct comparison of our two networks against other reported networks on the sMNIST, pMNIST, and sCIFAR-10 datasets. We reported the other networks in this table because we wanted to show the accuracies that are achievable on these datasets by using other methods. There are several reasons why we cannot do a direct comparison: the IndRNN network is a multi-layered architecture and we could not reproduce the results in the BN-LSTM work even though we used the reported training parameters.  We do not claim that the g-LSTM network can produce state-of-the-art results, rather we see it as an alternate model that helps long sequence training and could probably be used in addition to other methods such as the auxiliary loss in the r-LSTM work and IndRNN.

---

### Official Review · AnonReviewer3 · 2018-11-03
**Simple idea with potential but experiments are unconvincing**

**Rating:** 5
**Confidence:** 5

**Review:**

In the vein of recent work on learning “ticking” behaviour for LSTMs such as Phased LSTM, this paper proposes to add additional data independent gates to LSTM units that are defined as Gaussian functions of time indices.
The performance of the modified g-LSTM is compared to LSTM on the Addition, sequential MNIST and sequential CIFAR-10 tasks. The authors argue that g-LSTM results in better performance and has faster convergence on these tasks.

Additionally, it is proposed that one can reduce the amount of computations performed by the network by adding a computation budget term to the optimized loss that encouraged the cells to update less often. Finally, a technique for gradually transitioning from a g-LSTM to an LSTM during training is proposed, with the objective of speeding up training over a regular LSTM.

The paper is well written and easy to understand in general. However, the main results of this paper are experimental, and I am not entirely convinced by the experiments that g-LSTM is an improvement over the LSTM baseline for certain scenarios.

One broad reason for my doubts is that the comparisons don’t seem to utilise proper hyperparameter tuning for the baseline LSTM. Network sizes, learning rates, decay schedules, initialisations etc. all appear to be fixed, so one can not be sure of the “real” performance or convergence behavior of the models. Biased gate initializations are not used, though they have been used successfully in past work to aid in long term memory.

I should note that for long term memory problems such as those proposed by Hochreiter and Schmidhuber (1997), the proposed LSTM did not use a forget gate (or even BPTT) and used biased gate initialisations. However, these features are useful for more realistic tasks, and popular LSTM designs are biased towards them instead of toy problems.

I would consider the addition problem and sequential MNIST and CIFAR-10 to be interesting and difficult toy tasks for initial validation of ideas (and more extensive hyperparameter searches). It is unclear if the proposed techniques will perform provide  improvements over a well-tuned baseline for some realistic tasks, or are they suitable only for toy problems.

---

> ### Author Response · Authors · 2018-11-19
> **Response to AnonReviewer3**
>
> Thank you for your review.
>
> 1. We have rewritten the abstract, introduction and conclusion to make it clear that the advantages (convergence speed) of g-LSTM over the LSTM are prominent primarily in tasks where the sequence length is over 500,  whereas for shorter sequences we see that both the g-LSTM and LSTM networks perform similarly.
>
> 2. We have added more experiments over a range of hyperparameters. These results are reported in Appendix D. We see the benefit of using the g-LSTM across these additional experiments.
>
> 3. Regarding the baseline LSTM, we have taken care of optimizing hyperparameters and initializations. This is reflected in the performance of the baseline LSTM networks, which are on par with the performance of LSTM networks with similar network size in other publications, e.g. Recurrent Batch Normalization by Cooijmans et al.  We have also used biased gate initializations, which we have now added to the paragraph in Section 3.4.
>
> 4. We chose the datasets because they were used in several works that address the problem  of  long  sequence  training in recurrent  networks.  We agree with the reviewer that these datasets are “toy datasets” (useful to help develop new algorithms) and we are currently investigating the g-LSTM performance on practically relevant datasets.

---

> > ### Comment · AnonReviewer3 · 2018-11-26
> > **Response to authors' response**
> >
> > I have read the updates to the paper, and appreciate the changes that address some of my concerns. However, I am still not fully convinced that:
> >
> > a) the utility of biased initializations as a hyperparameter has been sufficiently explored or compared to for the baseline LSTM. My impression is that the authors are now using an initial bias of 1.0 for the forget gate(?) as done by Jozefowicz et al. which is a good start, but there are other good alternatives in past work. See Sec. 4.2 and 4.6 from Gers et al., as well as Tallec and Ollivier (ICLR 2018) for some good choices. These comparisons are important since the gate biases could play a similar role to the proposed input independent gates for synthetic long time lag problems by speeding up training.
> >
> > b) the absence of experiments on realistic benchmarks is okay in this case.
> >
> > I think that if the authors can clearly show that the proposed mechanism can provide benefits not possible to achieve by simply adjusting the initial biases of the gates, and that these benefits can be obtained on some realistic benchmarks as well, this will be a very nice study. Unfortunately for now, the results are still preliminary to me.
> >
> > Tallec, Corentin, and Yann Ollivier. "Can recurrent neural networks warp time?." ICLR 2018.
> > Gers, Felix A., Jürgen A. Schmidhuber, and Fred A. Cummins. "Learning to Forget: Continual Prediction with LSTM." Neural Computation 12.10 (2000): 2451-2471.

---

> > > ### Author Response · Authors · 2018-11-26
> > > **Follow up response**
> > >
> > > We appreciate your follow up response and have made some changes to the manuscript.
> > >
> > > a) We did use an initial bias of 1.0 for the forget gate in all the experiments in the paper. We have now carried out new experiments using the chrono initialization (of the bias) method as described in Tallec and Ollivier, 2018. The results are now added to Appendix D. We show that the g-LSTM still outperforms the LSTM initialized in such a fashion.
> > >
> > > b) From our preliminary results on Penn Tree Bank and text8, we do not observe any obvious advantages of the g-LSTM over LSTM besides faster convergence, similar to what was observed in (Tallec and Ollivier, 2018). We believe that this is because the sequences are not very long.

---

> > > > ### Comment · AnonReviewer3 · 2018-11-27
> > > > **2nd update review**
> > > >
> > > > I thank the authors for providing additional interesting results.
> > > >
> > > > a) The new experiment results in Appendix D indicate that initializing LSTM biases in a well motivated way already goes a long way towards closing the gap to g-LSTM (compare Fig. 12 and Fig. 2c). Results in Fig. 12 show that all comparisons in the paper should be done at least to LSTM+Chrono-init, and that Figure 2 in the main paper which is part of the main claims does not tell the full story.
> > > >
> > > > For the additional problem for example chrono init practically removed the initial flat region of training curve in past work (for up to T=750), so it is likely that it will help for the addition tasks here as well. Additionally, the impact of using a separate (higher) learning rate for the biases should be investigated.
> > > > Overall, it is not clear that concrete new insights can be drawn yet from the performance of g-LSTM, so my rating is unchanged.
> > > >
> > > > Further, there are even more ways of improving performance on certain synthetic tasks such as not using the forget gate altogether and not using full BPTT. For these reasons it is important to be careful when focusing solely on synthetic tasks.
> > > >
> > > > b) Faster convergence is a difficult claim to make in general, since one has to optimize hyperparameters for both target performance and convergence speed. Nevertheless, I would think that consistent faster convergence if rigorously demonstrated could be a nice positive.

---

### Author Response · Authors · 2018-11-19
**Manuscript Update**

We thank the reviewers for their insightful comments. We have rewritten the abstract, introduction, and conclusion to highlight more specifically when the g-LSTM will converge faster than the LSTM. We also added a new section (Section 5) to explain the faster convergence of g-LSTM and we have removed Fig. 2 from the original submission to make space for this new section.

---

### Author Response · Authors · 2018-12-09
**Additional author response**

We appreciate the knowledgeable and insightful comments regarding convergence. We acknowledge that there is substantial relevant work that shows improvement of convergence properties with various methods (bias initialization, gate/kernel initialization, auxiliary losses, learning rate scheduling);  however, with this paper we aim to highlight a new time-gated RNN mechanism which demonstrates faster convergence with reduction in state updates and which provides additional advantages.
We would like to reiterate the other important result of the paper: The time gate allows for a straightforward method towards incorporating a secondary, budget loss term - and we demonstrate that this additional term during training can be useful for reducing inference operations and even helps to prune the network while maintaining network accuracy.

---

### Meta-Review · Area_Chair1 · 2018-12-14
**reviewers are not convinced (nor am i)**

**Confidence:** 4
**Recommendation:** Reject

**Metareview:**

perhaps the biggest issue with the proposed approach is that the proposed approach, which supposedly addresses the issue of capturing long-term dependency with a faster convergence, was only tested on problems with largely fixed length. with the proposed k_n gate being defined as a gaussian with a single mean (per unit?) and variance, it is important and interesting to know how this network would cope with examples of vastly varying lengths. in addition, r3 made good points about comparison against conventional LSTM and how it should be done with careful hyperparameter tuning and based on conventional known setups.

this submission will be greatly strengthened with more experiments using a better set of benchmarks and by more carefully placing its contribution w.r.t. other recent advances.